# Perfectionistic Environments and Irrational Beliefs on the Transition to Elite Athletic Performance: A Longitudinal Study

**DOI:** 10.3390/ijerph20085561

**Published:** 2023-04-18

**Authors:** Yago Ramis, Joan Pons, Saul Alcaraz, Susana Pallares, Carme Viladrich, Juan Muñoz-Justicia, Miquel Torregrossa

**Affiliations:** 1Departament de Psicologia Bàsica, Evolutiva i de l’Educació, Universitat Autònoma de Barcelona, 08193 Bellaterra, Spain; 2Sport Research Institute, Universitat Autònoma de Barcelona, 08193 Bellaterra, Spain; 3Departament de Psicologia, Universitat de les Illes Balears, 07122 Palma, Spain; 4Departament de Psicologia Social, Universitat Autònoma de Barcelona, 08193 Bellaterra, Spain; 5Departament de Psicobiologia i de Metodologia de les Ciències de la Salut, Universitat Autònoma de Barcelona, 08193 Bellaterra, Spain

**Keywords:** perfectionism, irrational beliefs, motivation, transitions, youth sport, dual career

## Abstract

This study aimed to longitudinally evaluate talented athletes’ levels of perfectionism, irrational beliefs, and motivations with regard to their athletic careers. A total of 390 athletes from U14, U16, and junior categories (*M_ageT_*_1_ = 15.42) answered shortened versions of the Sport-MPS2, iPBI, and BRSQ during two consecutive seasons, along with questions referring to their current and predicted prioritization of sports and education. Participants reported high levels of perfectionistic strivings and medium to low levels of socially prescribed perfectionism and concern over mistakes decreasing from T1 to T2. A decrease was also found for demandingness and awfulizing, but increased levels were found for depreciation in T2. Participants report very high intrinsic motivation with low levels of external regulation and amotivation, but intrinsic motivation decreased from season to season. This general profile varied depending upon future expectancies toward dedication to sports and education. Those who foresaw a prioritized dedication to sports presented significantly higher levels of socially prescribed perfectionism, perfectionistic strivings, and intrinsic motivation, while those who considered that sports would not be prioritized in the following 5 years reported higher levels of demandingness, awfulizing, depreciation, and amotivation. Additionally, while current levels of motivation (T2) seemed to be predicted mainly by previous motivation levels (T1), significant predictive capacity was also detected for socially prescribed perfectionism positively predicting external regulations and amotivation, perfectionistic strivings negatively predicting amotivation, and depreciation negatively predicting intrinsic motivation and positively predicting both extrinsic regulation and amotivation. We discuss the potential perils of developing extremely demanding environments, as they could potentially result in poor motivational profiles of athletes in their talent development stage during the junior to senior transition.

## 1. Introduction

Thousands of young athletes worldwide practice a sport regularly, and a high percentage of them dream of the possibility of becoming professional athletes. However, the odds of attaining such an achievement are low. Based on previous literature, we know that less than 20% of young, talented athletes will successfully enter elite sports levels [1], and this figure drops even lower for popular sports, such as football [2,3,4]. In a recent paper by Jordana et al. (2022) [5], it was found that, in talent selection environments (such as football academies), more than 57% of the players mistakenly believe they will, in fact, become sports professionals. While this belief could boost their commitment in the first place, this can become a source of frustration and potentially threaten the wellbeing and mental health of junior athletes [6]. The dominant narrative of success in a sport, which is retrospective and reported only by those athletes who actually achieved top-level positions in elite sports, creates a dangerous atmosphere for talent development environments [7]. In such environments, considerations of success refer only to reaching professional sports, while moving to non-professional sports or dropping out is considered a failure [5,8]. Previous literature in the field refers to a perfectionism paradox in elite sports [9,10] that considers that, even though achieving elite performance and perfection is an almost unrealistic, dysfunctional, or irrational goal, the pursuit of perfect performance is set as a standard for athletes aiming to become elite professionals within sports.

The junior-to-senior transition (JST) is a critical process in career development and has been described as the most difficult transition faced by athletes [11]. Literature on this transition emphasizes the importance of personal characteristics (i.e., perceptions of the transition, motivation, and personal development) and social and environmental accompaniment (i.e., social support, motivational climate, and sources of stress) [12,13] to successfully cope with it. Career decisions, including the selection of career pathways and the prioritization of sports over other life spheres, greatly depend on the beliefs that athletes and their social environments have [14]. As irrational beliefs during this stage can result in dysfunctional emotions for athletes—thus risking their performance, wellbeing, and health—recent developments in the rational emotive behavior therapy (REBT) to sport approach [15] have focused on this critical JST period [5,16]. Irrational beliefs, such as demandingness, awfulizing, low frustration tolerance, and depreciation, have been related to greater perceptions of athletic competitions as more threatening and may, in some cases, result in less vitality as an indicator of well-being and mental health [17].

Originally defined as a form of irrational belief, perfectionism (i.e., setting excessively high standards for performance with overly critical evaluations of one’s behavior) [18], has recently been redefined to adapt to a more multidimensional interpretation that better matches the sport’s complexity [19]. This multidimensionality in perfectionism implies the split of (1) perfectionistic strivings, as a self-oriented effort toward excellence, organization, and setting high standards and (2) perfectionistic concerns, referring to worry about previous mistakes, a mismatch between one’s expectations and performance, and a fear of social evaluation [20]. A social perspective of perfectionism has also suggested an alternative dimensionality of the phenomenon, where socially prescribed perfectionism would refer to the conception that striving for perfection and being perfect are important to others, compared with self-oriented perfectionism, which would reference striving for perfection as personally important [21,22]. Perfectionistic concerns and socially prescribed perfectionism have been related to more ill-being consequences (e.g., anxiety, fear of failure, and performance-avoidance), while perfectionistic strivings can have positive consequences related to athletes’ wellbeing (e.g., positive emotions, self-confidence, and hope for success and performance) [19].

Self-determination theory (SDT) [23] considers that these wellbeing and ill-being indicators are linked to motivational processes but are also modulated by environmental factors. While this theory considers human beings as responsible for their own personal growth, integrity, and wellbeing, individuals’ environments and social agents are potential facilitators of or barriers to their basic psychological needs; thus, these factors partially determine their types of motivation. According to SDT, behavioral regulations are distributed along a continuum from more self-determined motivation (i.e., intrinsic motivation) to externally regulated behavior (i.e., extrinsic motivation) or non-regulation (i.e., amotivation)—with more self-determined regulations predicting wellbeing, commitment, and performance in comparison to non-regulated motivations, which are related to anxiety, depression, and low performance [24]. In fact, self-determined motivation can also have a mediational effect between the forms of perfectionism and maladaptive consequences, such as burnout and sport dropout [25,26]. These authors found that self-oriented perfectionistic strivings, combined with intrinsic motivation, would act as a protective factor, while the negative effects of burnout would be boosted by socially prescribed perfectionism together with non-regulated forms of motivation.

If motivation is a determinant of favoring successful within-career transitions [12,13], career pathways and prioritization also seem to play a part in developing athletes’ self-determined motivations. Chamorro et al. [2] found that those football academy players considering sports to be more important than other life spheres would report less autonomous motivation and more amotivation than players presenting more balanced prioritization. These results add to the evidence suggesting that dual careers—defined as pathways balancing goals for both the athletic and academic or vocational spheres—potentially have more beneficial consequences, both during and after the athletic career, in comparison to sport-only linear trajectories [27,28].

While sports psychology advocates for healthier and more humane sports conditions, the preparation required to achieve high performance is an extremely challenging process for potential elite athletes. In that sense, this work aimed to longitudinally assess the perceived levels of perfectionism, irrational beliefs, and behavioral regulations of young athletes on their path to high performance during two consecutive seasons. We assessed the flows from the past, current, and future prioritization of sports and education by talented athletes. Additionally, we compared athletes based on this prioritization, evaluating predictors of their current motivation toward sports and assessing differences in perfectionism, irrational beliefs, and motivational profiles.

## 2. Materials and Methods

### 2.1. Participants

A total of 390 talented athletes between 12 and 20 years of age (*M_t_*_1_ = 15.42; *M_t_*_2_ = 16.42; *SD* = 1.85) participated in this study, of which, 29% were female athletes. Only educational categories were included (i.e., U14, U16, and junior), excluding all participants who had already been part of the senior squads as an amateur, semi-professional, or professional. The criteria for considering participants to be talented athletes were that (a) the club they competed for had a competitive orientation, and (b) they played in talent-oriented categories within the club instead of recreation-oriented categories. The sports included were football, basketball, field hockey, and roller hockey. In terms of competitive level, 38.5% of the sample had participated in national or international competitions, while the complementary 61.5% had only competed in regional or inter-regional competitions. Participants were approached twice during two consecutive seasons, and demographic data, such as birthdate, club, sport, and initials representing first and last name allowed for the matching of cases from Time 1 (T1) to Time 2 (T2).

### 2.2. Measures

#### 2.2.1. Preparation Phase

This study used shortened versions of questionnaires assessing sport perfectionism, irrational beliefs, and behavioral regulation, following the recommendations of Alcaraz et al. [29]. Preliminary internal consistency tests were performed, suggesting more parsimonious and consistent structures for the factors of the MPS-2, the iPBI, and the BRSQ-2 questionnaires. Each adaptation is specified within the presentation of the measures. 

#### 2.2.2. Current and Future Dedication to Sports

To describe the reality of our participants, in terms of their prioritization of sports and studies, talented athletes were asked about their current and future balance between sports and academic spheres. The present items were worded as “Currently…” and the answer choices were “I’m dedicated exclusively to sport/I prioritize sports over studies/I have a similar commitment to both sports and studies/I prioritize studies over sports/I am dedicated exclusively to studies”. For the foresight of future dedication, the stem of the items was reworded to “In the future (next 5 years) I foresee myself…” with equivalent answer choices.

#### 2.2.3. Perfectionism

Sport perfectionism was assessed using the Sport Multidimensional Perfectionism Scale-2 (Sport MPS-2) [30] under the original translation of Ramis et al. [31] Based on the preparation phase of the measures, in the case of the MPS-2, we detected that items from the factors of perceived parental pressure and perceived coach pressure operated on a unitary basis within, what previous studies have defined as, socially prescribed perfectionism [26]. We erased items that referred to participants’ interpretations of other people’s actions (e.g., PPP2: I have the feeling that I could never achieve my family’s expectancies) to avoid misinterpretation of the factor. Accordingly, the Sport MPS-2 included 17 items grouped into 3 dimensions: socially prescribed perfectionism, which included 5 original items from the perceived parental pressure (e.g., “My family expects excellence from me in my sports career”) and perceived coach pressure (e.g., “My coach sets very high goals for me in my sport”) subscales; personal standards (8 items; e.g., “I set more demanding goals than most athletes in my sport”); and concern over mistakes (4 items; e.g., “The fewer mistakes I make in sport, the more people will accept me”). The internal consistency of the three factors of sport perfectionism was acceptable, at both T1 and T2, for socially prescribed perfectionism (*α_t_*_1_ = 0.77; *α_t_*_2_ = 0.70), perfectionistic strivings (*α_t_*_1_ = 0.81; *α_t_*_2_ = 0.80), and concern over mistakes (*α_t_*_1_ = 0.73; *α_t_*_2_ = 0.72). 

#### 2.2.4. Irrational Beliefs

To measure participants’ irrational beliefs regarding sports, items from the iPBI-2 were used and adapted [5,32]. Adapted specifically for performance settings, the iPBI-2 scale is a 16-item tool. For the iPBI, the factors of demandingness and awfulizing presented strong correlations, as previously noted in the original design of the inventory [30]. In our sample, as interpreted from the network analysis, they also behaved as a unitary factor, showing better internal consistency when combined than each of the conforming factors did independently. Therefore, they were analyzed jointly in the manuscript. Thus, the included factors are demandingness and awfulizing (8 items; e.g., “I have to be viewed favorably by people that matter to me”), low frustration tolerance (4 items; “I can’t bear not getting better at what I do”), and depreciation (4 items; “If others think I am no good at what I do, it shows I am worthless”). Internal consistency of the scale was acceptable, at both T1 and T2, for demandingness and awfulizing (*α_t_*_1_ = 0.74; *α_t_*_2_ = 0.74), low frustration tolerance (*α_t_*_1_ = 0.68; *α_t_*_2_ = 0.71), and depreciation (*α_t_*_1_ = 0.82; *α_t_*_2_ = 0.83).

#### 2.2.5. Motivation

Motivation for sports practice was assessed using the Behavioral Regulation in Sport Questionnaire (BRSQ2) [33], which was validated in Spanish by Viladrich et al. [34] The original version of the questionnaire evaluates all behavioral regulations of the self-determination continuum; however, for the purposes of this study, only three factors—evaluating intrinsic motivation (4 items; “I play sports because it’s fun”), external regulation (3 items; “I play sports because people push me to do so”), and amotivation (4 items; “I play sports, but I wonder why I keep doing it”)—were used. In this case, based on the preparation phase, we erased item 4 of the external regulation factor (i.e., I play sports to succeed, be rich and famous), as the item referred to material prices instead of other persons’ expectations, thus negatively affecting the internal consistency of the factor. Internal consistency of the three scales was acceptable along T1 and T2 for intrinsic motivation (*α_t_*_1_ = 0.79; *α_t_*_2_ = 0.87), external regulation (*α_t_*_1_ = 0.72; *α_t_*_2_ = 0.74), and amotivation (*α_t_*_1_ = 0.85; *α_t_*_2_ = 0.82).

### 2.3. Procedure

This study is framed within a bigger project, entitled “Healthy Dual Careers in Sport” (HeDuCa: RTI2018-095468-B-100), which is aimed at, among other goals, promoting the dual careers of talented athletes through the evaluation of psychological variables during their junior-to-senior transition. In order to assess the maximum number of variables in the most efficient manner, a shortening procedure was conducted with all questionnaires, following the recommendations of previous authors [35,36]. The selection of the key factors within the targeted questionnaires and the reduction of the number of items per factor was performed by a group of experts to gain greater information with not-so-demanding data collection procedures. Details can be reviewed in Alcaraz et al. [29].

Once we gained approval from the University’s ethics committee (ref. 4996), we developed the agreed-upon battery of questionnaires in the LimeSurvey online platform. Following purposeful sampling, we approached high-performance centers, federations, and clubs in order to get access to talented athletes within different sports, and we informed participants of our intention to follow up on their evolution during three consecutive seasons. Once permission was granted, we contacted coaches to distribute informed consent documents and agree to the dates for data collection from their teams. Questionnaires were administered under the in-person supervision of one or two researchers at the locations agreed upon by the sports organization. All collections were scheduled for between 20 to 30 min before training sessions. Participants were informed about the aims of the project, their voluntary participation, and the confidentiality of data usage. Participants could answer the questionnaires using their mobile phones. Data were collected without any important incidents, and all participants could follow their regular training routines after completing the questionnaires.

### 2.4. Data Analytic Strategy

As introduced in the preparation phase of the measures, internal consistency was tested using Cronbach’s alpha coefficient together with network analysis testing based on the “qgraph” package for R [37] on its JASP version v.0.17.1. This graphical representation allowed for the structuration of scales in the most parsimonious and reliable form, which also helped with the detection of problematic items on the shortened versions of the scales. The internal consistency of the scales was tested based on Cronbach’s alpha internal consistency test in SPSS 23.0. For the analysis of career prioritization, we used a Sankey diagram generated using the Flourish online application (app.flourish.studio, accessed on 29 December 2022). This diagram is a descriptive chart that allows for the analysis of flows between sample groups in a population across time, thus allowing for the presentation of the evolution of the prioritization spheres within our population. Quantitative analyses were conducted using the 23.0 version of IBM SPSS. Student’s *t*-test for repeated measures, with a confidence interval of 95%, was generated to compare T1 and T2 variables. In addition, to evaluate differences between the three groups of prioritization, a multivariate ANOVA was performed. Finally, to assess the main cognitive predictors of the current (i.e., T2) levels of each of the targeted variables of motivation (i.e., intrinsic motivation, external regulation, and amotivation), three stepwise multiple regression analyses were conducted that included all T1 variables and the T2 cognitive variables (i.e., perfectionism and irrational beliefs). 

## 3. Results

### 3.1. Past, Current, and Predicted Future Career Prioritization

A visual description of the evolution of career prioritization of talented athletes between seasons, along with the foresight of their predicted future dedication to sports and education, is presented in Figure 1. In our sample, 30% of participants (*n* = 115) were already more dedicated to sports than to education at T1, 16% were more dedicated to education than to sports (*n* = 63), and 52% (*n* = 204) considered sports and education to be balanced in prioritization. At T2, one season later, the participants choosing to prioritize sports increased to 34% (*n* = 133), of which, 4.1% (*n* = 16) reported an exclusive dedication to sports; the percentage of athletes prioritizing education also increased to 20% (*n* = 78); thus, those equally balancing sports and education decreased to 44% (*n* = 171).

With regard to future dedication predictions, the number of participants who believed that they would be able to make a living from sports exclusively increased to 13% (*n* = 51), which, when added to those who believed that sports will be prioritized, reaches 43% (*n* = 167), thus marking the biggest group in the distribution. Those who believed that education will be prioritized also increased to 25% (*n* = 98), with 2.5% (*n* = 10) anticipating a total dropout from sports. The group with the bigger decrease was the one who believed that sports and education will be balanced, 29% (*n* = 115). In terms of priority changes, 28% (*n* = 111) predicted that sports would have more dedication in their lives in the future, while 24% (*n* = 98) anticipated that sports would be less important in their lives in the future.

### 3.2. Perfectionism, Irrational Beliefs, and Motivation Overtime

In Table 1, we present a descriptive overview and comparison of the reported levels of perfectionism, irrational beliefs, and motivation for T1 and T2. Considering that the potential responses ranged from 1 to 7, the levels of perfectionism are situated at the mid-point of the scale. Socially prescribed perfectionism was below this midpoint in T1 (*M_t_*_1_ = 3.39) with a significant decrease in T2 (*M_t_*_2_ = 3.20), slightly higher levels were reported for perfectionistic strivings in T1 (*M_t_*_1_ = 4.62), and T2 (*M_t_*_2_ = 4.51) also presented a significant decrease over time. Lower levels were reported for concerns over mistakes in T1 and T2 (*M_t_*_1_ = 2.92; *M_t_*_2_ = 3.07)—in this case, presenting a significant increase from season to season. 

Regarding irrational beliefs, both the scales of demandingness and awfulizing (*M_t_*_1_ = 4.85; *M_t_*_2_
*=* 4.74) and low frustration tolerance (*M_t_*_1_ = 5.24; *M_t_*_2_
*=* 5.26) presented levels over the midpoint; only the first one presented a significant decrease over time. A different trend appeared for depreciation, with lower levels, in general, but presenting a significant increase from T1 (*M_t_*_1_ = 2.57) to T2 (*M_t_*_2_ = 2.72). Motivation levels presented a ceiling effect for intrinsic motivation, with a significant decrease from T1 (*M_t_*_1_ = 6.60) to T2 (*M_t_*_2_
*=* 6.49), and floor effects for external regulation (*M_t_*_1_ = 1.97; *M_t_*_2_
*=* 1.90) and amotivation (*M_t_*_1_ = 1.95; *M_t_*_2_
*=* 1.96), without significant changes over time. 

### 3.3. Analysis of Variance with Regard to Future Career Expectancies

To evaluate differences between groups at T2 based on their future career expectancies, we conducted an ANOVA with three groups defined as follows: prioritizing sports, balanced prioritization, and sports not prioritized (see Table 2). For variables of perfectionism, differences were found for socially prescribed perfectionism (*F* = 13.74; *p* < 0.001) and perfectionistic strivings (*F =* 43.82; *p* < 0.001) but not for concern over mistakes. Specifically, the group prioritizing sports had significantly higher levels of socially prescribed perfectionism (*M* = 3.52) and perfectionistic strivings (*M* = 5.00) than those who would balance sports and education (*M* = 3.13; *M* = 4.43) and those who will not prioritize sports (*M* = 2.76; *M* = 3.84). For irrational beliefs, significant differences were also found for demandingness and awfulizing (*F =* 8.62; *p* < 0.001) and depreciation (*F =* 3.16; *p* = 0.044), with the group prioritizing sports presenting significantly lower levels of demandingness and awfulizing (*M* = 4.55) in comparison with both the group of balanced prioritization (*M* = 4.81) and the group with sports not prioritized (*M* = 4.97), and lower levels of depreciation (*M* = 2.60) in comparison to the group with sports not prioritized (*M* = 2.97). Finally, significant differences were also found in current motivation levels for the three groups in intrinsic motivation (*F* = 4.36; *p* = 0.013) and amotivation (*F* = 12.03; *p* < 0.001), with the group with sports not prioritized presenting lower levels of intrinsic motivation (*M* = 6.29) than the group prioritizing sports (*M* = 6.59) and higher levels of amotivation (*M* = 2.42) than the group with balanced prioritization (*M* = 1.96) and the group prioritizing sports (*M* = 1.67).

### 3.4. Regression Analysis for Current Motivation Levels

Targeting T2 motivation as the dependent variable, we conducted a multivariate stepwise regression analysis considering behavioral regulations (T1) and perfectionism and irrational beliefs (T1 and T2) as potential predictive variables (see Table 3). 

Additionally, we generated change variables calculating the variations from T1 to T2 for perfectionism and irrational beliefs to assess their potential predictive capacity for current behavioral regulations.

Predictably, current intrinsic motivation was mainly predicted, positively and significantly, by T1 intrinsic motivation (*t* = 8.52; *p* < 0.001) in the first place, but also by a negative change in depreciation (*t* = −4.59; *p* < 0.001) and by a positive change in demandingness and awfulizing (*t* = 2.55; *p* = 0.011). This implies that increasing depreciation and decreasing demandingness and awfulizing from season to season predicts lower levels of intrinsic motivation. T1 levels of concern over mistakes (*t* = −2.73; *p* = 0.007) and T2 levels of low frustration tolerance (*t* = 2.54; *p* = 0.012) also presented negative and positive predictive capacity, respectively, over intrinsic motivation.

Current levels of external regulation were also mainly predicted by the T1 levels of this form of regulation (*t* = 5.86; *p* < 0.001), but almost with an equivalent predictive capacity to that of current levels of socially prescribed perfectionism (*t* = 5.73; *p* < 0.001). Again, the increase of depreciation between T1 and T2 predicted external regulation (*t* = 3.82; *p* < 0.001). Negative predictive effects were also found for T1 intrinsic motivation (*t* = −2.91; *p* = 0.004) and T2 low frustration tolerance (*t* = −2.51; *p* = 0.012).

Finally, the current levels of amotivation were positively predicted by T1 levels of this form of non-regulation (*t* = 7.00; *p* < 0.001), but also by T2 current levels of depreciation (*t* = 6.13; *p* < 0.001). T1 intrinsic motivation negatively predicted current levels of amotivation (*t* = −2.97; *p* = 0.003). Changes in specific variables also had predictive effects on amotivation, with significant negative effects of increased perfectionistic strivings (*t* = −3.55; *p* < 0.001) and demandingness and awfulizing (*t* = −2.50; *p* = 0.013). On the other hand, increased socially prescribed perfectionism positively predicted current levels of amotivation (*t* = 2.09; *p* = 0.037).

## 4. Discussion

This manuscript has longitudinally evaluated levels of perfectionism, irrational beliefs, and motivations among talented athletes on their path toward top sports competition levels. Our proposal informs how perfectionism and rationality, together with career expectancies, predict the current motivational profiles of talented athletes. Results suggest that environmental conditions and dispositional characteristics might have predictive utility in anticipating the wellbeing or ill-being of talented athletes on their JST.

The profile of athletes in our sample is that of players who are striving for perfect performance and consider that their environment partially reinforces these high standards. They present low frustration tolerance levels, in general, and even lower levels of irrational depreciation of themselves. The group is highly intrinsically motivated toward sports, presenting low levels of external regulation and amotivation. In terms of what comes from the evolution of the prioritization of sports and education, our sample shows profiles that are increasingly drifting away from a balanced dual career logic, with one growing group of athletes who believe they will be able to live out their lives within professional sports, and another group considering that sports will fall into the background of a career mainly focused on education. This indicates that, even though literature provides evidence in favor of dual careers to protect healthy athletic trajectories and smoother retirement processes [28,38], in this initial stage of career prioritization, athletes considered sports an all-or-nothing project. 

Increasing numbers of athletes willing to dedicate themselves exclusively to sports in the near future, who present perfectionistic profiles with higher levels of external demands (socially prescribed perfectionism) and internal self-imposed standards (perfectionistic strivings), combined with the fact that worse adaptive rational and motivational profiles are found among those who realize that sports will not be prioritized, presents a worrying scenario [5]. Failure to meet the expectations to reach elite sports—which would be the case for approximately 75% of the talented athletes in our sample—could, in fact, be associated with thousands of self-depreciated and amotivated youngsters who have devoted their time and efforts to sports, as previously warned in literature [8].

Our results complement previous data on junior-to-senior transitions in male soccer players [5] while adding evidence from different sports and including female athletes in the analysis. In our study, participants who consider that sports will not be prioritized in their future, report worse motivational profiles toward sports (i.e., lower intrinsic motivation and higher amotivation) but also present higher levels of demandingness, awfulizing, and depreciation. Such a profile could indicate that those athletes that begin to give up the possibility of dedicating themselves professionally to sports, based on current performance, would be developing what previous authors have labeled as contingent self-worth (i.e., “I’ve failed, therefore I’m a failure”) [30] when comparing the irrational beliefs of amateur and semi-professional athletes.

The results of our study reinforce the importance of the quality of motivation in youth sports, indicating that current motivational profiles and environments are highly influential in future motivations for sports [39]. Expectably, previous levels (T1) of intrinsic motivation, external regulation, and amotivation positively predicted current levels (T2) of the same regulations. However, intrinsic motivation also negatively predicted the development of external regulation and amotivation. This indicates that enjoyment and legitimate interest in sports act as protective factors against the appearance of external regulations or non-regulated behavior. Beyond these within-construct predictions, irrational beliefs have also proved their predictive capacity for motivational forms. The increase of self-depreciation from T1 to T2 has appeared as one of the main predictors of lower intrinsic motivation and higher external regulation, and momentary levels of this irrational self-evaluation also predicted general amotivation. The opposite happened with demandingness and awfulizing, with mild positive predictive capacity over intrinsic motivation and negative predictive capacity over amotivation. These relationships between irrational beliefs and motivations suggest that, while primary beliefs of demandingness might be adaptive in the transition to elite performance, depreciation clearly harms a healthy JST period [17]. Socially prescribed perfectionism—defined as perfectionistic demands from family and coaches—had a significantly predictive, positive effect on external regulation and, to a lower degree, on the levels of amotivation among talented athletes. In contrast, increased levels of perfectionistic strivings—a form of self-oriented perfectionism—negatively predicted levels of amotivation, reinforcing the conception of an adaptive effect of perfectionism when it is self-oriented, in contrast to an increased social burden when pressures to be perfect are perceived as originating from the environment [22,40].

The results of our study described high-demanding athletic environments in which athletes, coaches, and families share a drive to achieve high athletic performance. The self-oriented perfectionism and demandingness of the athletes seemed to facilitate more intrinsically motivated profiles, thus facilitating the continuation of athletes in the pursuit of that challenging goal. However, when athletes perceived that perfectionism was socially prescribed—mainly when it included thoughts of self-depreciation—this motivation worsened notably with increases in external regulations or even non-regulations of their participation, which is a key indicator of potential drop-out from sports [41]. Additionally, these dysfunctional profiles seemed to be more frequent among those athletes who plan for, or at least presume that, that sport will not be prioritized in their future, thus suggesting that anticipated deselection can have strong negative consequences on the motivation and wellbeing of talented athletes [42]. This fact suggests that it is reasonable to hold the environments responsible for accompanying athletes, even—or especially—when they are not achieving high-performance levels. This will, in fact, be the case in approximately 75–80% of the cases, thus redounding to the evidence from previous studies holding that coaches’ and families’ leadership and motivational climates can determine motivation and its consequences in youth sports [43,44,45].

This research is integrated into a greater project entitled “Healthy Dual Careers in Sport (HeDuCa: RTI2018-095468-B-100)”, and the versions of the questionnaires were edited to guarantee short and agile data collection. However, based on these edits, we had to concede including all of the factors in the BRSQ-2, thus attaining a clearer definition of the motivational profiles of the talented athletes in our study was not possible. Further studies, willing to draw more complete motivational profiles, should take this into consideration. In addition, and despite the longitudinal design of our study, including a longer follow-up period for our participants would have allowed for a clearer correspondence between perfectionistic levels, irrational beliefs, and motivational profiles during the complete JST period or, probably, during the deselection process. In our study, we could only infer participants’ continuation in high-performance sports based on their own predictions; however, the authors of this study are currently designing a follow-up, mixed-methods approach to generate more consistent results in regard to these variables.

## 5. Conclusions

Talented athletes, on their path to elite sports, find themselves in extremely demanding environments and entourages expecting perfect performance from them. While self-oriented demandingness and perfectionism seem adaptative for developing more intrinsic motivation to cope with the junior-to-senior transition, external pressures and self-depreciation might fuel worse motivational profiles, thus anticipating ill-being indicators and potential dropout from sports. Prospects of dedication to sports seem to be associated with more adaptative rational and motivational indicators, but responsibility needs to be put on athletic environments to accompany and protect, especially, those who will not follow a high-performance athletic career.

## Figures and Tables

**Figure 1 ijerph-20-05561-f001:**
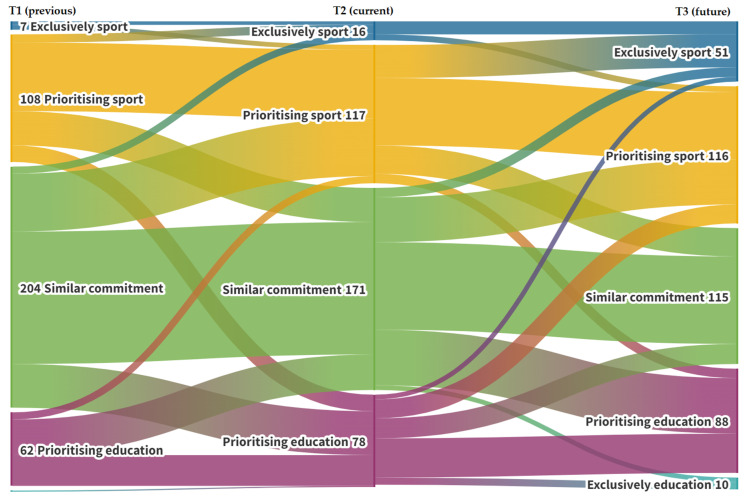
Sankey diagram of the evolution of prioritization for sports and education among talented athletes. Note: T1, time 1; T2, time 2; T3, time 3.

**Table 1 ijerph-20-05561-t001:** Descriptive statistics and mean comparisons for the variables of irrational beliefs, perfectionism, and motivation for time 1 and time 2.

	Time 1		Time 2		Diff.	
	*M* (*SD*)	*α*	*M* (*SD)*	*α*	*t*	*p*
Perfectionism						
Socially prescribed perfectionism	3.39 (1.35)	0.77	3.20 (1.20)	0.70	−3.06 **	0.002
Perfectionistic strivings	4.62 (1.15)	0.81	4.51 (1.09)	0.80	−1.99 *	0.047
Concern over mistakes	2.92 (1.28)	0.73	3.07 (1.25)	0.72	2.65 **	0.008
Irrational beliefs						
Demandingness and awfulizing	4.85 (0.91)	0.74	4.74 (0.86)	0.74	−2.39 *	0.017
Low frustration tolerance	5.24 (1.05)	0.68	5.26 (0.96)	0.71	0.30	0.761
Depreciation	2.57 (1.27)	0.82	2.72 (1.25)	0.83	2.41 *	0.016
Motivation						
Intrinsic motivation	6.60 (0.68)	0.79	6.49 (0.82)	0.87	−2.79 **	0.006
External regulation	1.97 (1.27)	0.72	1.90 (1.17)	0.74	−0.66	0.512
Amotivation	1.95 (1.35)	0.85	1.96 (1.25)	0.82	0.45	0.653

Note: All scales had a potential response range of 1–7; *M*, mean; *SD*, standard deviation; *α*, Cronbach’s alpha coefficient; Diff., difference testing. * = *p* < 0.05; ** = *p* < 0.01.

**Table 2 ijerph-20-05561-t002:** Mean comparison of T2 perfectionism, irrational beliefs, and motivation of talented athletes based on future career expectancies.

	Prioritizing Sports *n* = 163 *M* (*SD*)	Balanced Prioritization *n* = 115 *M* (*SD*)	Sports Not Prioritized *n* = 103 *M* (*SD*)	*F*	*p*
Perfectionism					
Socially prescribed perfectionism	3.52 (1.12) ^BP,SNP^	3.13 (1.23) ^PS^	2.76 (1.15) ^PS^	13.74	<0.001
Perfectionistic strivings	5.00 (0.97) ^BP,SNP^	4.43 (0.96) ^PS,SNP^	3.84 (1.04) ^PS,BP^	43.82	<0.001
Concern over mistakes	3.11 (1.27)	3.06 (1.19)	3.02 (1.28)	0.19	0.830
Irrational beliefs					
Demandingness and awfulizing	4.55 (0.79) ^BP,SNP^	4.81 (0.94) ^PS^	4.97 (0.80) ^PS^	8.62	<0.001
Low frustration tolerance	5.35 (0.92)	5.23 (1.05)	5.14 (0.95)	1.49	0.226
Depreciation	2.60 (1.19) ^SNP^	2.65 (1.23)	2.97 (1.34) ^PS^	3.16	0.044
Motivation					
Intrinsic motivation	6.59 (0.74) ^SNP^	6.53 (0.81)	6.29 (0.93) ^PS^	4.36	0.013
External regulation	1.82 (1.05)	2.06 (1.33)	1.86 (1.16)	1.56	0.212
Amotivation	1.67 (1.00) ^SNP^	1.96 (1.33) ^SNP^	2.42 (1.37) ^PS,BP^	12.03	<0.001

Note: *M*, mean; *SD*, standard deviation; PS, prioritizing sports; BP, balanced prioritization; SNP, sports not prioritized.

**Table 3 ijerph-20-05561-t003:** Regression analyses of current behavioral regulation among talented athletes.

Variable	*B*	*β*	*SE*	*t*	*p*	95% CI
	Intrinsic Motivation					
Constant	2.90		0.43	6.74	<0.001	[2.06, 3.75]
Intrinsic motivation T1	0.50	0.40	0.06	8.52	<0.001	[0.39, 0.62]
Δ Depreciation	−0.15	−0.22	0.03	−4.59	<0.001	[−0.21, −0.09]
Δ Demandingness and awfulizing	0.12	0.13	0.05	2.55	0.011	[0.03, 0.21]
Concern over mistakes T1	−0.09	−0.13	0.03	−2.73	0.007	[−0.15, −0.02]
Low frustration tolerance T2	0.11	0.12	0.04	2.54	0.012	[0.02, 0.19]
	External Regulation					
Constant	2.84		0.62	4.61	<0.001	[1.63, 4.06]
External regulation T1	0.27	0.29	0.05	5.86	<0.001	[0.18, 0.36]
Socially prescribed perf. T2	0.27	0.28	0.05	5.73	<0.001	[0.18, 0.37]
Δ Depreciation	0.16	0.17	0.04	3.82	<0.001	[0.08, 0.25]
Intrinsic motivation T1	−0.24	−0.14	0.08	−2.91	0.004	[−0.41, −0.08]
Low frustration tolerance T2	−0.14	−0.12	0.06	−2.51	0.012	[−0.26, −0.03]
	Amotivation					
Constant	3.17		0.70	4.53	<0.001	[1.80, 4.56]
Amotivation T1	0.32	0.34	0.05	7.00	<0.001	[0.23, 0.42]
Depreciation T2	0.28	0.28	0.05	6.13	<0.001	[0.19, 0.37]
Intrinsic motivation T1	−0. 28	−0.15	0.09	−2.97	0.003	[−0.46, −0.09]
Δ Perfectionistic strivings	−0.23	−0.17	0.06	−3.55	<0.001	[−0.35, −0.10]
Δ Demandingness and awfulizing	−0.16	−0.11	0.06	−2.50	0.013	[−0.28, −0.03]
Δ Socially prescribed perf.	0.11	0.10	0.05	2.09	0.037	[0.01, 0.22]

Note: Dependent variables presented refer to time 2 levels of intrinsic motivation, external regulation, and amotivation. T1, time 1; T2, time 2; Δ, change between time 1 and time 2; *SE*, standard error; CI, confidence interval.

## Data Availability

The data presented in this study are available on request from the corresponding author. The data are not publicly available due to the ongoing evaluation process of the funding institution.

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
