# Peer review of "Perfectionistic Environments and Irrational Beliefs on the Transition to Elite Athletic Performance: A Longitudinal Study"

_ijerph, 2023, doi:10.3390/ijerph20085561_

Round 1

Reviewer 1 Report

This manuscript aims to longitudinally assess the perceived levels of perfectionism, irrational beliefs and behavior regulations of young athletes on their paths to high performance during two consecutive seasons. I really enjoy reading this study. I appreciate the clarity of writing during the whole manuscript. However, I have two main concerns:

Participants of the study are named talented athletes. However, most of them compete in regional or inter-regional competitions only (61,5%). Why do authors consider them talented athletes or why they decided to consider the participants as talented?

The authors decided only to use 3 behavioral regulations of the self-determination continuum. Why? I think that using all of them gives us richer information. Or why not autonomous and controlled regulation? Or self-determination index? Theoretically, I think that anyone of these three options would be better, but I really want to read the justification of the authors.

Specific comments:

line 103. It misses the reference number of Chamorro et al.

line 346. It is not necessary to express the whole concept (Junior to Senior Transition) if previously in the introduction section you specified the abbreviation. 

Author Response

Dear Reviewer, first of all we want to thank you for the time dedicated to review our manuscript helping us to improve it in order to make it more rigorous and smooth for readers. We have tried to include modifications in order to meet most of your requirements and have answered the required procedural explanations.  

Comments and Suggestions for Authors

This manuscript aims to longitudinally assess the perceived levels of perfectionism, irrational beliefs and behavior regulations of young athletes on their paths to high performance during two consecutive seasons. I really enjoy reading this study. I appreciate the clarity of writing during the whole manuscript. However, I have two main concerns:

Participants of the study are named talented athletes. However, most of them compete in regional or inter-regional competitions only (61,5%). Why do authors consider them talented athletes or why they decided to consider the participants as talented?

Yes, the concept of talented athletes, as defined by Wylleman & Reints (2010) refers to athletes that are on their pathway to eventually become high-performance athletes. In that sense, we consider not to the actual level of competition but the characteristics of the environment in which they are developing and the team in which they are competing. We have included a modification in page 3 (L124-127) specifying the criteria used to consider these athletes talented athletes.

The authors decided only to use 3 behavioral regulations of the self-determination continuum. Why? I think that using all of them gives us richer information. Or why not autonomous and controlled regulation? Or self-determination index? Theoretically, I think that anyone of these three options would be better, but I really want to read the justification of the authors.

We totally agree that using all 6 original factors of the BRSQ would give much more detail on the motivational profiles of our sample. Unfortunately, the project in which this study is included involved several other questionnaires and we had to give up from using whole versions of some tools. We are aware that assessing controlled or autonomous motivation could present more balanced interpretation, but generating those factors would have prevent from assessing amotivation as separed from external regulation which we consider was an important nuance as amotivation refers to a lack of motives (and related with the intention to dropout) and external regulation refers to the pressures from others, which for the aims of the study was specially relevant. We disagree that assessing a self-determination index would have been more informative as the number of items to generate such and SDI are the same and the interpretation of such index has been criticized before (Chemolli & Gagné, 2014; Martín-Albo, González-Cutre, & Núñez, 2014; Wilson, Sabiston, Mack, & Blanchard, 2012).

Specific comments:

line 103. It misses the reference number of Chamorro et al.

Corrected

line 346. It is not necessary to express the whole concept (Junior to Senior Transition) if previously in the introduction section you specified the abbreviation. 

Corrected

Reviewer 2 Report

GENERAL COMMENTS

Dear Editor/Authors,

I would like to thank the editor for giving me the opportunity to review this manuscript. The topic is within the scope of the journal. The manuscript is well written and structured and reaches the research quality for publication.

I have found some minor issues that I detail in the specific comments above. Please find them constructively.

SPECIFIC COMMENTS

None of the authors is affiliated with Institution Number 4.

Abstract- The abstract is well structured and written.

Keywords- I would recommend authors to include the word youth or junior as in that way readers could easier find this article.

INTRODUCTION

The introduction is well written and structured.

L114-117 I find it somewhat complicated to understand the second aim or assessment. I suggest the authors to rewrite it in shorter sentences.

METHODS

L 120-123 The age of the sample (12-20 years) does not match the age level of competition (U12-U18) as those athletes with 19 or 20 years old will probably compete at senior/amateur/professional squads. Please check this issue.

L131-148 This subsection helps readers to clarify the measures. I would suggest that the authors include this information in the explanation of each tool.

L213-225 Please specify the software use for the network analysis testing and the Sankey diagram. Also inform about the number of regression analyses performed (3) and the set of alpha levels.

RESULTS

I have some suggestions for the tables.

Table 1 Insert * in the significant differences.

Table 3. The arrow may lead readers to think that there was an increase from T1 to T2. I would suggest authors to use a delta symbol if possible. Δ

DISCUSSION

L343-354 I find this sentence too long.

L 365 The authors did not measure the representativeness of their sample with the reference population, so please limit the writing to “our sample” or “the athletes from our study”…

Author Response

Dear Reviewer, first of all we want to thank you for the time and dedication to review our manuscript. Your comments and suggestions will help improve our proposal and make it more inviting for readers. We have answered to all of your comments below and included and highlighted modifications in the attachment.  

GENERAL COMMENTS

Dear Editor/Authors,

I would like to thank the editor for giving me the opportunity to review this manuscript. The topic is within the scope of the journal. The manuscript is well written and structured and reaches the research quality for publication.

I have found some minor issues that I detail in the specific comments above. Please find them constructively.

SPECIFIC COMMENTS

None of the authors is affiliated with Institution Number 4.

Thanks for noticing. Corrected.

Abstract- The abstract is well structured and written.

Keywords- I would recommend authors to include the word youth or junior as in that way readers could easier find this article.

We agree that this is an adequate keyword to include. We have now included “youth sport” instead of “sport”.

INTRODUCTION

The introduction is well written and structured.

L114-117 I find it somewhat complicated to understand the second aim or assessment. I suggest the authors to rewrite it in shorter sentences.

We agree that this final sentence was not easy to read. We have now split this in two sentences.

METHODS

L 120-123 The age of the sample (12-20 years) does not match the age level of competition (U12-U18) as those athletes with 19 or 20 years old will probably compete at senior/amateur/professional squads. Please check this issue.

We understand that this could be confusing. In fact we avoid to refer to U18 in our manuscript as this is not true for junior categories in some sports (e.g., football). We refer to U14, U16 and Junior instead. Some of the players in junior categories were already 19 years-old but none of them reached 20. We have corrected this in L121.

L131-148 This subsection helps readers to clarify the measures. I would suggest that the authors include this information in the explanation of each tool.

Thanks for the suggestion. We have kept the general explanation of the measure preparation phase but specified the actual modifications of each questionnaire on their corresponding subsection. Modifications can be found in L138, L152-157, L170-175 and L188-191.

L213-225 Please specify the software use for the network analysis testing and the Sankey diagram. Also inform about the number of regression analyses performed (3) and the set of alpha levels.

Specification of the R software and the corresponding package to perform the network analysis is now included in P219, we include the Cronbach’s alpha test specifying the use of SPSS in P222, and we specify the online app that allowed the generation of the Sankey diagram.

RESULTS

I have some suggestions for the tables.

Table 1 Insert * in the significant differences.

Done.

Table 3. The arrow may lead readers to think that there was an increase from T1 to T2. I would suggest authors to use a delta symbol if possible. Δ

We agree that the delta is more adequate that the arrow. We have now included this modification.

DISCUSSION

L343-354 I find this sentence too long.

Agreed. We have now split this sentence in two on L351-354.

L 365 The authors did not measure the representativeness of their sample with the reference population, so please limit the writing to “our sample” or “the athletes from our study”…

We understand that this sentence could be misleading and have modified this on L373.

Round 2

Reviewer 1 Report

Thank you for the responses provided.